# Conformational Insights into the Control of CNF1 Toxin Activity by Peptidyl-Prolyl Isomerization: A Molecular Dynamics Perspective

**DOI:** 10.3390/ijms221810129

**Published:** 2021-09-20

**Authors:** Eléa Paillares, Maud Marechal, Léa Swistak, Landry Tsoumtsa Meda, Emmanuel Lemichez, Thérèse E. Malliavin

**Affiliations:** 1Unité des Toxines Bactériennes, UMR CNRS 2001, Institut Pasteur, 75015 Paris, France; elea.paillares@pasteur.fr (E.P.); maud.marechal@pasteur.fr (M.M.); lea.swistak@pasteur.fr (L.S.); landry.tsoumtsa-meda@pasteur.fr (L.T.M.); 2Université de Paris, Sorbonne Paris Cité, 75006 Paris, France; 3Unité de Bioinformatique Structurale, UMR CNRS 3528, Institut Pasteur, 75015 Paris, France; 4Centre de Bioinformatique, Biostatistique et Biologie Intégrative, USR CNRS 3756, Institut Pasteur, 75015 Paris, France

**Keywords:** *Escherichia coli*, CNF1, deamidase, X-Pro imide bond, peptidyl prolyl *cis*-*trans* isomerization, molecular dynamics, peptide docking

## Abstract

The cytotoxic necrotizing factor 1 (CNF1) toxin from uropathogenic *Escherichia coli* constitutively activates Rho GTPases by catalyzing the deamidation of a critical glutamine residue located in the switch II (SWII). In crystallographic structures of the CNF1 catalytic domain (CNF1^CD^), surface-exposed P768 and P968 peptidyl-prolyl imide bonds (X-Pro) adopt an unusual *cis* conformation. Here, we show that mutation of each proline residue into glycine abrogates CNF1^CD^ in vitro deamidase activity, while mutant forms of CNF1 remain functional on RhoA in cells. Using molecular dynamics simulations coupled to protein-peptide docking, we highlight the long-distance impact of peptidyl-prolyl *cis*-*trans* isomerization on the network of interactions between the loops bordering the entrance of the catalytic cleft. The energetically favorable isomerization of P768 compared with P968, induces an enlargement of loop L1 that fosters the invasion of CNF1^CD^ catalytic cleft by a peptide encompassing SWII of RhoA. The connection of the P968 *cis* isomer to the catalytic cysteine C866 via a ladder of stacking interactions is alleviated along the *cis-trans* isomerization. Finally, the *cis-trans* conversion of P768 favors a switch of the thiol side chain of C866 from a resting to an active orientation. The long-distance impact of peptidyl-prolyl *cis*-*trans* isomerizations is expected to have implications for target modification.

## 1. Introduction

Uropathogenic *Escherichia coli* (UPEC) are the causative agents of at least 80% of urinary tract infections affecting 150 million people annually and worldwide [1]. The cytotoxic necrotizing factor 1 (CNF1) toxin encoding gene displays a prevalence of around 30% in UPEC, to which it confers high capacities to invade epithelial cells and to trigger inflammatory mediator production [2,3,4,5]. CNF1 belongs to AB-type protein toxins from pathogenic bacteria. The B-subunit is responsible for the binding of the toxin to a host cell surface-exposed receptor and internalization of the toxin-receptor complex into endocytic vesicles and trafficking up to specific compartments from which the B-subunit triggers the injection of the A-subunit into the cytosol. Once it reaches the cytosol, the A-subunit catalyzes an irreversible posttranslational modification (PTM) of essential host cell regulators. The CNF1 toxin catalyzes the specific deamidation of a critical glutamine residue of the small Rho GTPases (Q63 in RhoA and Q61 in Rac1 and Cdc42) into glutamic acid [6,7,8]. This posttranslational modification inhibits the GTPase activity thereby rendering Rho proteins permanently activated. The observation of isolated structures of CNF1 catalytic domain (CNF1^CD^) and RhoA indicates that major conformational changes are required to impose a close proximity of the thiol-group of the amino acid residue of Cys 866 of CNF1^CD^ responsible for the nucleophilic attack and the targeted δ-carboxamide of RhoA Q63 [9]. Rigid body docking does not allow to bring close enough C866 to Q63, as the entrance of the CNF1 active site is too shallow. Furthermore, we expect large conformational reorganizations not only of CNF1 but also of RhoA, as the SWII loop of RhoA has to move apart from the core of the GTPase structure to invade CNF1 catalytic pocket.

The planar peptide bond (C-N) linking two adjacent amino acids in a protein adopt either a *cis* or *trans* conformation, although it is known to occur predominantly in the *trans* configuration [10,11]. For all amino acid residues except proline the trans conformation is far more energetically favorable. Peptide bond *cis*-*trans* isomerization achieves large conformational changes without modifying the covalent structure of the proteins [10]. This consists of a switch of the backbone dihedral angle in X-Pro imide bonds between 180° (*trans*) and 0° (*cis*). Statistical analysis of protein structures demonstrated that *trans*-conformers are more prevalent and are more energetically favorable than *cis* conformers [11]. The free energy barrier to achieve their isomerization is on the order of 20 kcal/mol for X-Pro imide bond, a value lower than that for other amino acid amide bonds. The X-Pro *cis* conformers in the protein data bank are more prevalent than any other X-amino acid bonds [12,13] and peptidyl prolyl *cis*-*trans* isomerization can play a key role in protein function [14]. The importance of these modifications has mainly been established by studying peptidyl-prolyl *cis*-*trans* isomerase (PPIase) functions due to technical limitations in direct detection by biochemical characterization of peptidyl-prolyl imide bond *cis-trans* states. Nevertheless, seminal studies have uncovered that isomerization of peptidyl-prolyl imide bond increases the flexibility of proteins, thereby controlling essential cellular processes, including cell signaling, channel gating and gene regulation [10,15,16]. PPIase activities have also been diverted by the Apicomplexan parasites *Theileria parva* or *Theileria annulata* to promote leukocyte transformation and by *Legionella pneumophila* to facilitate macrophage infection [17,18]. While host PPIases promote the translocation of several bacterial AB toxins through host endosomal membranes during their translocation into the cytosol, there is still no evidence that such enzymes are diverted by toxins to ensure a specific posttranslational modification of their target once the A-subunit reaches the cytosol [19].

AB-like toxins act frequently in a concerted action with host cofactors and cosubstrates and by other refined molecular mechanisms to increase spatiotemporal specificity of their action once inside the cytosol. Such refined molecular mechanisms of action have not yet been described for CNF1. Here, using several in silico approaches, we have studied the impact of P968 and P768 peptidyl-prolyl isomerization on structural elements of the catalytic domain of the bacterial toxin CNF1. In the following, CNF1 denotes the multi-domain toxin, whereas CNF1^CD^ denotes the catalytic domain of CNF1. This analysis has been grounded by the observation that both X-Pro imide bonds of surface-exposed P768 and P968 display a *cis* conformation in the available X-ray crystallographic structures of CNF1^CD^ wild-type and C866S catalytic inactive mutant [9]. Based on molecular dynamics (MD) simulations, we have explored the effects of peptidyl-prolyl isomerization, providing a total of 1.6 μs of trajectories. To investigate the interaction between CNF1^CD^ and RhoA in the most objective way, we coupled molecular dynamics simulations of CNF1^CD^ to sample conformations with the use of FlexPepDock [20] and to dock on CNF1^CD^ the SWII peptide of RhoA and the equivalent SWII of the small GTPase Ras, as negative control. This analytical strategy allows to consider the possibilities of conformational changes of the two partners.

## 2. Results

### 2.1. Essential Roles of P768 and P968 in CNF1-Mediated Deamidation of RhoA In Vitro

Entering this study, we were intrigued by the presence of two *cis* isomers of surface-exposed X-Pro imide bonds in the structures of the enzymatic domain of the CNF1 toxin WT and C866S [9]. The deamidase domain CNF1^CD^ forms a compact structure composed of two mixed β-sheets arranged in a central sandwich surrounded by short α-helices and a network of loops extending at 10 Å over the catalytic site [9], as depicted in the Figure 1A. This network of loops organizes a narrow and deep catalytic cavity. Consistent with the observed confined size of the catalytic cleft, the specific calculated constant K_cat_/K_m_ of CNF1^CD^ for RhoA displays a modest value [21]. Several loops control the rate of RhoA deamidation and the specificity of modification of Rho proteins among other small GTPases of the Ras superfamily [21,22]. Loop-8 (L8) in CNF1^CD^ and proline 968 (P968) in L8 are critical elements in the proper deamidation of RhoA in vitro [21,22]. Paradoxically, the mutant P968Q of CNF1 is devoid of activity on RhoA in vitro but shows deamidase activity toward RhoA in intoxicated cells [22]. The lack of co-crystallization of the complex formed between CNF1 and RhoA has largely blurred our understanding of the molecular determinants controlling the specificity of the reaction toward the subfamilly of Rho GTPases.

Sequence conservation analysis among CNF1, 2 and 3 toxins from *Escherichia coli* identified a proline residue at the same position in the L8 of CNF2 (Figure 1B). Moreover, CNF3 and CNFy from *Yersinia pseudotuberculosis* also display a conserved proline residue in L8, although this proline residue is shifted by two amino acids with respect to its location in the CNF1 and CNF2 sequences. Proline 768, which lies in L1 between β-sheets 1 and 2 (Figure 1A), is strictly conserved among these CNF members (Figure 1B). The difference in sequence conservation between P768 and P968 is probably a sign for a difference in the relative role of these prolines in the toxin function. To investigate this, we mutated each proline residue into a glycine residue, and we examined the impact of these mutations on the deamidase activity of CNF1^CD^ toward RhoA in vitro. Deamidation of RhoA can be monitored based on a visible electrophoresis mobility shift of its deamidated form. Both CNF1^CD^ P768G and P968G displayed impaired deamidase activities despite long time periods of reaction (Figure 1C). In contrast, mutations P768G and P968G did not abrogate the effect of CNF1 on RhoA in intoxicated cells, where CNF1 effect is monitored by analyzing the proteasomal degradation of RhoA (Figure 1D). This behavior of CNF1 P968G is in accordance with previous data showing that the mutation P968Q rather increases CNF1 activity on RhoA in cells [22]. Moreover, the activity of CNF1 proline mutants on RhoA in cells argues against an absence of activity observed in vitro due to alterations of the structure.

In conclusion, P768 in loop-1 and P968 in loop-8 are both essential to CNF1 deamidase activity in vitro, while these mutant forms of CNF1 are active on RhoA in cells.

### 2.2. X-Pro Imide Bond cis-trans Isomerization and Tertiary Structure of CNF1^CD^

We took advantage of molecular dynamics approaches to model the peptidyl-prolyl isomerizations of P768 and P968 in CNF1^CD^. Different force constants were applied for restraints on the ω angle of P768 and/or P968 X-Pro imide bonds (Table 1). At smaller force constants, we observed that P768 switches to the *trans* isomer after 10 ns in trajectory tr_10 (Figure 2B, black curves), whereas P968 took up to 60 ns to switch from the *cis* to the *trans* isomer for tr968_10 and tr_10 (Figure 2E, black and green curves). Imposing higher force constants of 100 kcal/(mol.degrees) switched both prolines to *trans* isomers during the initial preparation steps of the MD simulations; thus, the restrained prolines were *trans* isomers during the entire MD trajectory (Figure 2C,F). We then examined the overall conservation of CNF1^CD^ conformations. The coordinate root-mean-square deviations (RMSDs) displayed a plateau at approximately 2 Å for all trajectories (Appendix A). This average distance between atoms establishes that the global structure of CNF1^CD^ remained unaffected along molecular dynamics trajectories. Similarly, no significant variation in the secondary structure of CNF1^CD^ was observed using the define secondary structure of proteins (DSSP) method [25] (data not shown).

We analyzed the stability of the sandwich formed by the two mixed β-sheets by measuring the distances between the heavy backbone atoms of β-strands facing each other in opposite β-sheets. Variations in these distances were approximately 0.2 Å in most cases, corresponding to a decrease of approximately 2% with respect to the original distances between β-sheets (Appendix A). The largest variations (on the order of 0.3–0.4 Å) were observed at the extremities of the two mixed β-sheets for couples β-1/β-3 and β-10/β-13. Together, measurements of the distances between heavy backbone atoms established the strong conservation of the structure of CNF1^CD^ while X-Pro imide bonds undergo *cis-trans* conversion.

### 2.3. X-Pro Imide Bond cis-trans Isomerization Affects the Organization of the Network of Loops Restricting Catalytic Cavity Accessibility

We analyzed the interactions between amino acid residues in the loops located at the entrance of the catalytic groove of CNF1^CD^. In the proline *trans*-isomer model, loops L1, L2, L3, L6, L7, L8 and L9 display relatively large coordinate root-mean-square fluctuations (RMSFs) (Å) (Figure 3A) and RMSDs (Å) (Figure 3B) with respect to the initial conformation in 1HQ0. Hydrogen bonds and van der Waals interactions were then monitored along the trajectories to investigate the conformational variations of the loops more precisely and to connect them to changes in *trans* isomerization of proline residues. Analysis of hydrogen bonds as described in [26], showed that several interloop distances displayed trends toward larger values associated with a disruption of corresponding hydrogen bonds. Figure 4 illustrates the rupture of the connection between the amide hydrogen HN from E943 (L7) and the Oδ atoms of D965 (L8), between the Hγ1 atom from S836 (L4) and the Oδ atoms of D789 (L2), and between the carbonyl oxygen from S941 (L7) and the amide hydrogen HN from D965 (L8). In contrast, the distance between the Hγ1 atom from T885 (L6) and the Oδ atoms of D789 (L2) decreased upon *trans* isomerization, leading to the formation of a hydrogen bond. These modest numbers of atom pairs displaying distance variations between the loops connecting β-strands agree with the overall rigidity observed when the architecture of the core structure of the mixed β-sheet sandwich was analyzed. Importantly, these data show an increase in the fluctuations of loops bordering the catalytic cavity following *trans* isomerization of P768 and P968.

We noticed in the 1HQ0 structure that proline P968 (L8) is connected to the catalytic residue C866 through a ladder of stacking interactions between aromatic residues (Figure 5A, left panel). Residue P968 is stacked on residue phenylalanine F963, which in turn is stacked on tyrosine residue Y962. Although Proline is not an aromatic residue, stacking interactions between prolines and aromatic amino acids have been energetically described [27]. In addition to stacking interactions, the hydroxyl group of tyrosine Y962 establishes a hydrogen bond with the carbonyl oxygen of the catalytic residue C866. We monitored the variation in stacking for F963/P968 and for F963/Y962 (Figure 5A, right panel) by calculating the averaged distances between the atoms of the aromatic or aliphatic cycles. Variations in the F963/P938 and F963/Y962 stacking distances had fairly different trends between the *cis* and *trans* conformers. Residues F963, Y962 and P968 displayed stable stacking interactions when both proline residues were in *cis* (Figure 5B, top panel) or when P768 or P968 adopted a *trans* conformation (Figure 5B, magenta, or green contours in the bottom panel). In contrast, the ladder of stacking interactions was destabilized once the two proline residues underwent *trans* isomerization (Figure 5B, black contour in the bottom panel).

### 2.4. Accessibility to the Catalytic Cavity of CNF1^CD^

We then explored the global impact of peptidyl-prolyl isomerization on the accessibility of catalytic residues within the catalytic pocket. The region encompassing the deeper part of the cavity appeared unaffected along MD trajectories, as the accessible surfaces (Å^2^) of C866 and H881 varied little from the small value of 10 Å^2^ while the accessible surface of the third residue of the catalytic triad, V833, is negligible (Appendix A). Nevertheless, the outlier’s values of the surface of C866 are larger than for the H881 surface. Together, these data show that residues C866, H881 and V833 display similar accessibility regardless of the restraints applied on the X-Pro imide bonds of P768 and P968.

We then analyzed the accessibility of the catalytic cavity of CNF1^CD^ at three increasing distances from the catalytic residues. To this aim, the number of water molecules bordering CNF1 catalytic cleft in the 90–100 ns interval of the trajectories was averaged for three spheres with radii of 15, 20 and 25 Å that were centered on C866 (Appendix A). Notably, for the 90–100 ns time interval of each trajectory, the proline residues effectively switched toward a *trans* conformation. We found that restraints imposing a *trans* conformation on P768 and/or P968 induced slightly larger dispersion in the number of water molecules (Appendix A, green, magenta, orange boxes). The dispersion is more obvious for spheres with a radius of 25 Å but also slightly visible for spheres of smaller radii. In good agreement with the fluctuations in the network of loops bordering the entrance of the catalytic groove, our analysis shows that the upper part of the catalytic pocket displays wider variations than the lower part.

### 2.5. Docking of the RhoA SWII Target Peptide to CNF1^CD^ cis-trans Isomer Models

We decided to exploit molecular dynamics simulations of CNF1 *cis-trans* isomers to compare the docking efficiency of the switch II domain of RhoA between the different isomers. The docking was performed on the representative conformations of CNF1^CD^ extracted from the trajectories by the self-organizing map approach, as described in the Appendix A. We then selected a peptide of RhoA (Table 2) that is known to be deamidated in vitro and includes both residues R68 and L72, which discriminate RhoA from other small GTPases of the p21-Ras superfamily [16,28]. The equivalent peptide of p21-Ha-Ras was thus used as a negative control (Table 2). Next, FlexPepDock software was run to dock the two peptides on the various *cis-trans* isomers of CNF1^CD^ [21]. During the docking procedure, no specific restraints between CNF1 ^CD^ and the peptides were applied to avoid bias in the simulations. We pooled all docking solutions together, as described in the methods section. Briefly, the docking solutions were selected if the distance between the Hγ atom of CNF1 C866 and the N2 peptide atom of RhoA Q63 was less than 5 Å and the distance between the Cα atoms of residues 60 and 72 of the peptide was less than 15 Å. Remarkably, the percentages of solutions were greater for RhoA than for Ha-Ras in all conditions (Figure 6). This points to a specific interaction between the SWII domain of RhoA and CNF1^CD^. In addition, we observed an increase of success rate only for the conformations of CNF1^CD^ extracted from tr768_10 and tr768_100 trajectories (1.38 × 10^−4^ and 1.84 × 10^−4^), as compared to tr968_10 and tr968_100 (0.64 × 10^−4^ and 0.62 × 10^−4^) (Figure 6). Thus, despite a *trans*-isomerization of P968 that is observed in tr968_100 the success rate of interaction of RhoA SWII display control values. Moreover, the recorded rates were consistently greater for P768 *trans* isomers (1.85 × 10^−4^) than for *cis* (0.64 × 10^−4^) and free (0.48 × 10^−4^) cases. Taken together, these data indicate that when P768 adopts a *trans* conformation the catalytic cleft of CNF1^CD^ is more amenable to invasion by the SWII domain of RhoA, a phenomenon that is not observed when P968 adopts a *trans* conformation.

The analysis of docking solutions obtained for the SWII domain of RhoA revealed a larger number of hydrogen bonds between RhoA residues for the docking realized on tr768_10 and tr768_100 conformations than for docking performed using conformations from other trajectories (Appendix A). Consequently, a greater number of stable β hairpins are observed in RhoA for docking solutions in these trajectories. Moreover, we also observed a larger number of hydrogen bonds between SWII domain and CNF1^CD^ residues for these docking experiments (Appendix A).

Closer analysis of MD conformations revealed that hydrogen bonds for residues N762/Y772 and for D764/F770 within CNF1^CD^ were destabilized as soon as P768 or P968 X-Pro imide bonds adopted *trans* conformations (Figure 7B, magenta and green contours). The *trans* isomerization of both X-Pro imide bonds stabilized the hydrogen bonds. Refined analysis shows that the disruption of the N762/Y772 and D764/F770 hydrogen bonds between backbone atoms separates the β1 and β2 strands in the CNF1^CD^ structure (Figure 7A), thereby enlarging L1. This enhances the accessibility of the loops bordering the entrance of the catalytic site, which aligns with the recorded increase in interactions between the RhoA SWII domain and CNF1 ^CD^ (Appendix A). A similar model has been proposed for the interleukin-2 tyrosine kinase SH2 domain, where prolyl isomerization of the prolyl imide bond N286-P287, located in the so-called CD loop, mediates conformer-specific ligand recognition [29]. The variation of conformation for loop L1 in CNF1^CD^ plays a role equivalent to variation of conformation for loop CD in interleukin-2.

### 2.6. Orientation of the Thiol Side Chain of the Catalytic Cysteine C866

In the crystal structure 1HQ0, the side chain of C866 displays an equilibrium between two orientations: a resting orientation (dihedral angle χ1 (N-Cα-Cβ-Sγ) of −63.5°, g- rotamer) and an active orientation (dihedral angle of 40.6°). In the resting orientation, the thiol side chain of C866 points toward H881 and is thus not available to participate in the reaction, whereas in the active orientation, this group is perpendicular to the plane formed by the H881 sidechain and is thus free for the reaction (Figure 8A). Distributions of the χ1 angle of the C866 SH group were extracted from the trajectories in which P768 or P968 occupied a defined *cis* or *trans* conformation to tentatively relate the configuration of the CNF1 active site to proline *cis-trans* conformers (Figure 8B). The χ1 angle distribution displayed two maxima in all cases, corresponding to resting (−63.5°) and active (60°: g+ rotamer) orientations (Figure 8B). According to the proline *cis-trans* status, we measured different amplitudes for the two maxima. For trajectories in which proline residues are *cis* (unbroken curve), the active orientation was twice as populated as the resting state. The ratio increased when one or two proline residues adopted a *trans* conformation (dashed curve). Thus, enforcing the transition of one of the proline residues into a *trans* conformation shifts the thiol side chain of C866 toward an orientation advantageous for catalysis.

## 3. Discussion

We report that the proline residues P768 (in loop-1, L1) and P968 (in loop-8, L8) are essential to CNF1 deamidase activity measured in vitro, thereby unveiling the unexplored role of loop-1 in CNF1 enzymatic activity. Our in silico studies show that peptidyl prolyl *cis*-*trans* isomerization of P768 and P968, while it has no impact on the overall structure of CNF1^CD^, increases the flexibility of the network of loops at the entrance of the catalytic cleft. We found that this also controls the switch of the thiol reactive group of cysteine 866 from a resting to an active orientation and the capacity of the switch II domain of RhoA to invade the catalytic cleft of the CNF1^CD^ more efficiently when P768 adopts a *trans*-conformation. Moreover, we found that peptidyl prolyl *cis-trans* isomerization of P968 specifically destabilizes its connection to the catalytic cysteine residue C866 but does not improve the interactions between CNF1 and the SWII domain of RhoA. Finally, P768 displays a lower energetic barrier for *cis*-*trans* isomerization. Thus, all our data show the control of the state of activation of the CNF1 catalytic domain via a X-Pro imide bond isomerization-based mechanism of P768.

Based on extensive molecular dynamics simulations, recent work showed spontaneous *cis-trans* isomerization in the CDR-H3 loop of an antibody in response to antigen binding [30]. The authors described the transition as being supported from changes in the interactions between residues of the loop. Glycine and polar residues were reported as important in that context. Notably, in the case of CNF1^CD^, the presence of several polar residues (D764, E765, Q766, and Q767 in L1 and D965, N966, and E969 in L8) and glycine residues in L8 (G971 and G973) gives a context similar to that observed in [30]. Nevertheless, from the in silico analysis presented here, it is obvious that the *cis-trans* peptidyl prolyl isomerization of P768 and P968 are not equivalent. From a kinetics point of view, the *cis*-*trans* transition of P968 is achieved at a higher energy than that of P768. This finding is consistent with the observation that the environments of the proline residues are different. P768 is relatively accessible and isolated from the core structure of CNF1, whereas P968 is surrounded by many residues in the initial conformation and is connected to the catalytic cysteine residue.

We found that the upper part of the CNF1 catalytic cavity displays a greater range of conformations as soon as one or both prolyl bonds switch to *trans*-conformers. The consequences of *cis-trans* isomerization of X-Pro imide bonds were further investigated using a peptide docking strategy. Notably, the facilitation of peptide docking is most likely underestimated here. In the FlexPepDock strategy, the protein is kept rigid, and the peptide-induced fit on CNF1 is then neglected. Moreover, the docking calculation only considers the association step between CNF1^CD^ and the peptides without considering variations in dissociation values. Nevertheless, refined analysis of the effects of peptidyl prolyl *cis*-*trans* isomerization on the interaction of the SWII domain of RhoA highlights the importance of the *trans* conversion of P768. The facilitation of RhoA docking is not observed after *trans* isomerization of P968.

Washington and collaborators have reviewed the accumulated knowledge on the structures of various deamidase effectors of bacteria alone or co-crystallized with their targets [31,32,33,34,35]. Their review highlighted various structural reorganization requirements aiming at restricting deamidase reactions in place and against highly specific targets, to which we propose to add the *cis-trans* isomerization of proline residues of CNF1-like toxins from *E. coli*. These requirements encompass a reduction of the disulfide bridge as a means to assemble a reactive catalytic site, most likely after the enzyme is exposed to the reducing conditions of the host cell cytosol. Analysis of the catalytic C3-domain of the PMT toxin from *Pasteurella multocida* shows the engagement of the critical cysteine residue C1165 in a disulfide bond with C1159 [31]. Mutation of C1159 to serine induces a displacement of C1165 toward the putative catalytic pocket. Similarly, the catalytic cysteine residue C62 of the OspI deamidase from *Shigella flexneri* is covalently bound to C65 via a disulfide bond [34]. The Cif-like virulence factors found in various *Enterobacteriaceae* provide us with a mode of tight adjustment between the deamidase catalytic pocket and their substrates NEDD8 ubiquitin-like molecule and ubiquitin [32,36]. Here, a so-called conserved occluding loop partially blocks the substrate binding site at the entrance of the catalytic site [31,35,36,37]. This occluding loop provokes a displacement of the flexible carboxy-terminal tail of ubiquitin and NEDD8, thereby allowing formation of the complex and orientation of the target glutamine residue toward the active site. There is also a reorganization of the catalytic pocket mediated by the interaction of the deamidase with its target in the case of OspI [34].

A recent study reported several crystallographic structures of the toxin CNFy from *Yersinia pseudotuberculosis* [38], including various combinations of CNFy domains. The structures containing the D5 domain corresponding to the catalytic domain (PDB entries: 6YHK, 6YHM, 6YHN) display features in agreement with the observations made here: the X-P970 imide bond in the loop L8 of CNFy displays a *cis* conformation in 6YHM, and a stacking and hydrogen bond network P970/L969/Y963/Y962/C866 similar to the network P968/F963/Y962/C866 observed in CNF1. The X-P768 imide bond displays a *trans* conformation in CNFy structures 6YHK, 6YHM and 6YHN along with a breaking of hydrogen bonds between D764 and D770, in agreement with the observations made here in the P768 *trans*-isomer of CNF1 for D764 and F770 and with the consequences of P768 isomerization on L1 enlargement. The X-P768 imide bond *cis-trans* conformations in the 3D structures of the catalytic domain of the two toxins may involve differences in barrier energy due to differences of primary sequence environment (QPL for CNF1 and RPG for CNFy) also indicating that the X-P768 imide bond equivalent in CNF-like toxins can adopt either one of the two *cis*-*trans* conformations.

In conclusion, from the observation of two X-proline imide bonds in a *cis* conformation in the structure of CNF1^CD^, we propose that the peptidyl prolyl *cis-trans* isomerization of proline 768 favors the invasion the catalytic cleft of CNF1 by the SWII domain of RhoA.

## 4. Materials and Methods

### 4.1. Preparation of WT in Silico System for MD Simulations

An X-ray crystallographic structure of CNF1^CD^ [9] (PDB entry: 1HQ0, 1A) was the starting point of the molecular dynamics (MD) trajectories. Once the sulfate ions were removed, chain A of the structure was analyzed using MolProbity [39] (molprobity.biochem.duke.edu) to add hydrogen atoms and to select the sidechain orientations optimizing the network of hydrogen bonds. At that stage, H881 was doubly protonated. The protein was then neutralized using one sodium ion and solvated with TIP3P water [40] molecules (Table 3). The force field CHARMM27 [41,42] was used to model physical interactions.

### 4.2. Restraints on X-Pro Imide Bonds

For the simulations, depending on the chosen conditions, the conformations of P768 and P968 were modified by applying an isomerization restraint on the dihedral angle *θ* between atoms O-(i−1), C-(i−1), N−i, and Cα−i, where i is the residue number of prolines. The restraint was applied along the potential U(*θ*) = k(1 + cos(n*θ* − *θ*_ref_)) where n = 1 and *θ*_ref_ = 180° for the *cis* restraint and *θ*_ref_ = 0° for the *trans* restraint [42]. The minimum value of U(*θ*) is obtained for the *θ* values of 180° and 0° of the angle ω of the peptide bond for the *cis* and *trans* restraint. The force constant *k* was equal to 10 or 100 kcal/(mol.degrees) depending on the trajectories (Table 1). The restraint was applied starting from the minimization step of the simulations.

### 4.3. Recording MD Trajectories

The MD trajectories were recorded using NAMD 2.7b2 [43]. The simulations were realized in the NPT ensemble at a temperature of 300 K and pressure of 1 atm. Temperature was regulated according to a Langevin thermostat [44], and the pressure was regulated with the Langevin piston Nose-Hoover method [45,46]. A cutoff of 12 Å and a switching distance of 10 Å were defined for nonbonded interactions, while long-range electrostatic interactions were calculated with the particle mesh Ewald (PME) protocol [47]. The RATTLE algorithm [48,49] was used to keep rigid all covalent bonds involving hydrogens, enabling a time step of 2 fs to be used. Atomic coordinates were saved every 10 ps. At the beginning of each trajectory, the system was first minimized for 1000 steps and then heated up gradually from 0 to 300 K in 30,000 integration steps. Finally, the system was equilibrated for 50,000 steps. For each simulated system, two independent trajectories of 100 ns were recorded, denoted replicas R1 and R2. Simulations were recorded in eight various conditions (Table 1), producing an overall trajectory of 1.6 μs

### 4.4. Analysis of MD Trajectories

The root-mean-square deviations (RMSDs, Å) and the root-mean-square fluctuations (RMSFs, Å) of atomic coordinates were calculated using cpptraj [24]. Distance and angle analysis along the recorded trajectories was also performed using cpptraj. The solvent-accessible surfaces of residues along the trajectory were calculated using a Python script based on the Python MDAnalysis library [50,51] coupled to the software FreeSASA [52].

### 4.5. Clustering of Conformations Sampled Along MD Trajectories

Representative conformations of a given trajectory were extracted using the self-organizing map (SOM) approach [53], which is described in the Appendix A. The size of the SOM map was 50 × 50, and the structure descriptors used as inputs were the distances between Cα atoms. The other input parameters are the same as those used in [54].

### 4.6. Peptide Docking

The interactions between CNF1^CD^ and 60–72 peptides of the G proteins RhoA and Ras were studied using FlexPepDock [20]. A flexible docking software was chosen as a large conformational variation of the peptide spanning the domain SWII of RhoA, given the narrow configuration of the CNF1 catalytic site. Extended conformations of 60–72 peptides (Table 2) were generated using the leap tool of AMBER [55]. Peptide docking was performed on each of the representative conformations of CNF1^CD^ conformations extracted with the SOM approach, using extended peptide conformations manually positioned in front of the catalytic pocket as starting points. Two opposite initial orientations of the peptide were used. Given the large number of docking trials performed in the range of 128,000 to 230,400 for the various trajectories, the relative small numbers of starting points for the peptides are not expected to induce any bias.

### 4.7. Selection of Peptide Conformations

Along the docking calculations, only the conformations in agreement with the enzymatic activity of CNF1 on RhoA were kept for further analysis. These conformations were selected based on the following criteria: (i) the distance between the Hγ atom of CNF1 C866 and the Nε2 peptide atom of Q63 undergoing deamidation (labeled with an asterisk in Table 2) was smaller than 5 Å and (ii) the distance between the Cα atoms of residues 60 and 72 of the peptide was smaller than 15 Å. The first criterion (i) was meant to provide a protein/peptide position agreeing with the deamidation reaction of Q63, whereas the second criterion (ii) was meant to obtain peptide conformations roughly fitting the folded structure of the G protein. The distance threshold for criterion (ii) was estimated from an analysis of the Cα-Cα distances between residues 60 and 72 in the RhoA structures corresponding to the following PDB entries: 1A2B [56], 1CC0 [57], 1CXZ [58], 1DPF [59], 1FTN [60], 1KMQ [61], 1LB1 [62], 1S1C [63], 1X86 [64], 1XCG [65], 2RGN [66], 3KZ1 [67], 3LW8 [68], 3LWN [68], 3LXR [68], 3MSX (to be published), 3T06 [69], 3TVD [70], 4D0N [71], 4F38 [72], 4XH9 [73], 4XOI [74], 4XSG [75], 4XSH [75], 5A0F [76], 5BWM [75], 5C2K (to be published), 5C4M (to be published), 5EZ6 (to be published), 5FR1 [77], 5FR2 [77], 5HPY [78], 5IRC [79], 5JCP [80], 5JHG (to be published), 5JHH (to be published), 6BC0 [81], 6BCA [68], and 6BCB [68]. This analysis revealed that the distances between the Cα atoms were in the range 9.4–10.8 Å.

### 4.8. Directed Mutagenesis and Modification of RhoA

The recombinant CNF1^CD^ encompassing amino acids 720–1014 was expressed using pGEX-2T and purified as described in [82]. The wild-type and mutant forms of His-tagged CNF1 toxin and GST-CNF1^CD^ were purified on HiTrap^TM^ TALON crude (Cytiva, Uppsala, Sweden) and GSTrap HP (GE Healthcare, Vélizy, France) respectively, as described by the manufacturer. The P768G and P968G mutant forms of CNF1 or CNF1^CD^ were obtained by site-directed mutagenesis of CCT into GGT using QuickChange Lighting (Agilent Technologies, Les Ulis, France), as described by the manufacturer. Reactions of deamidations were conducted at 37 °C in 100 mM NaCl, 50 mM Tris, pH 7.4 with recombinant GST-CNF1^CD^ WT and mutants 0.04 or 0.08 mg/mL together with recombinant RhoA (Sigma, St. Quentin Fallavier, France) 0.005 or 0.01 mg/mL, respectively. Aliquots were taken every 30 min, and the reactions were stopped by adding 2X Laemmli blue buffer (Sigma, St. Quentin Fallavier, France) at a 1:1 ratio. Samples were boiled at 100 °C for 5 min, and proteins were resolved on 15% SDS-PAGE (40% Acrylamide/Bis Solution 29:1, Bio-Rad, Marnes-la-Coquette, France) with 1 M Urea (Sigma, St. Quentin Fallavier, France). Gels were stained using Instant Blue Coomassie protein stain (Abcam, Cambridge, nited Kingdom). Intoxication experiments were conducted on Human Umbilical Vein Endothelial cell HUVECs (Promocell, Heidelberg, Germany), as described in [3]. Cells were intoxicated with 10nM of CNF1 WT and mutants. Cells were lysed in 2X Laemmli blue buffer (Sigma, St. Quentin Fallavier, France) at 1:1 ratio. Samples were boiled at 100 °C for 5 min, and proteins were resolved on 4–12% SDS-PAGE prior to transfer on PVDF membrane 0.45 μm (GE Healthcare, Vélizy, France) for immunoblotting. Immunoblots were performed with monoclonal anti-RhoA (#sc-418, Santa Cruz, Heidelberg, Germany) and anti-GAPDH (#sc-47724, Santa Cruz, Heidelberg, Germany) and polyclonal goat anti-mouse immunoglobulin HRP as secondary antibody (P0447, DAKO, Les Ulis, France). Signals were revealed with Immobion Western Chemiluminescent HRP Substrate (Merck, St. Quentin Fallavier, France) and recorded on PiXi (Syngene, Cambridge, UK).

## Figures and Tables

**Figure 1 ijms-22-10129-f001:**
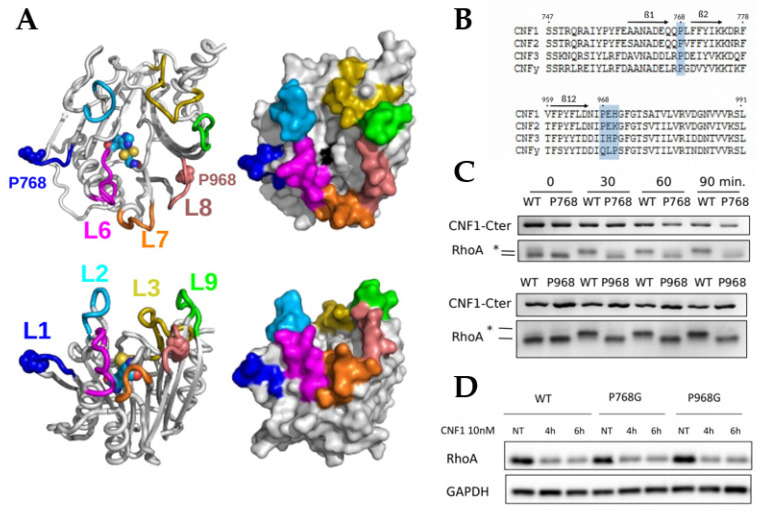
Sequence conservation and functional importance of P768 and P968 residues. (**A**) structure of CNF1^CD^. Overview (top) and side-view (bottom) of the X-ray crystallographic structure 1HQ0 [9]. The protein structure is drawn as a cartoon (left) and as a surface (right). The upper loops are colored blue (L1), cyan (L2), yellow (L3), violet (L6), orange (L7), pink (L8), and green (L9). Proline 768 (L1) and 968 (L8) and the catalytic residues C866 and H881 are drawn as spheres. C866 is colored black on the protein surface. The figures were generated using PyMOL [23]. (**B**) Conservation of P768 and P968 among CNF-like factors. Protein sequences of CNF1 (CAA50007), CNF2 (WP057108870), and CNF3 (WP024231387) from *E. coli* and CNFy (WP012304286) from *Yersinia pseudotuberculosis* were retrieved from NCBI. The alignment was generated using Clustal Omega [24]. The blue highlight indicates the positions of P768 and P968 in L1 and L8, respectively. (**C**) Altered deamidase activity of CNF1^CD^ mutants P768G and P968G in vitro. Kinetics of RhoA deamidation by CNF1^CD^ WT is visualized by monitoring the upper shift of RhoA on SDS-PAGE (*). Upper shift of RhoA is not observed with CNF1^CD^ P768G and P968G even after 90 min reaction. Figures show representative experiments, *n* = 3. (**D**) The targeting of cellular RhoA by CNF1 WT and mutants is monitored by following the toxin-induced proteasomal degradation of the GTPase. Immunoblots show efficient degradation of RhoA in cells intoxicated by CNF1 WT, P768G and P968G, one representative experiment, *n* = 3. Immunoblots anti-GAPDH show equal protein loading. HUVECs were left untreated or intoxicated 4 and 6 h with 10nM of CNF1 WT and mutants (NT: non-treated). Targeting of RhoA by CNF1 is monitored by following the toxin-induced proteasomal degradation of the GTPase.

**Figure 2 ijms-22-10129-f002:**
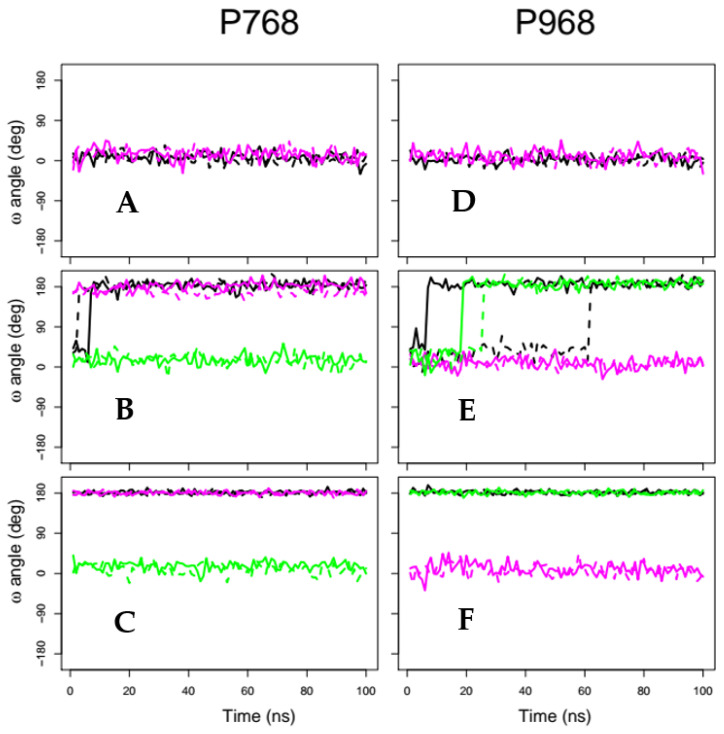
Global impact of X-Pro imide bond *cis-trans* isomerization. Variation in the X-Pro imide angle ω of P768 (**A**–**C**) and P968 (**D**–**F**) along MD trajectories. Solid and dashed curves correspond to replicas R1 and R2 of the same trajectory. The curves are colored in the following way: (**A**,**D**) trajectories with *cis* X-Pro imide bonds: cis (black), free (magenta); (**B**,**E**) trajectories with isomerization restraints using a force constant of 10 kcal/(mol.deg) tr_10 (black), tr768_10 (magenta), and tr968_10 (green) (several transitions of P768 and P968 are visible during the trajectories); (**C**,**F**) trajectories with isomerization restraints using a force constant of 100 kcal/(mol.deg): tr_100 (black), tr768_100 (magenta), and tr968_100 (green). For clarity, only one point in every 100 frames is plotted.

**Figure 3 ijms-22-10129-f003:**
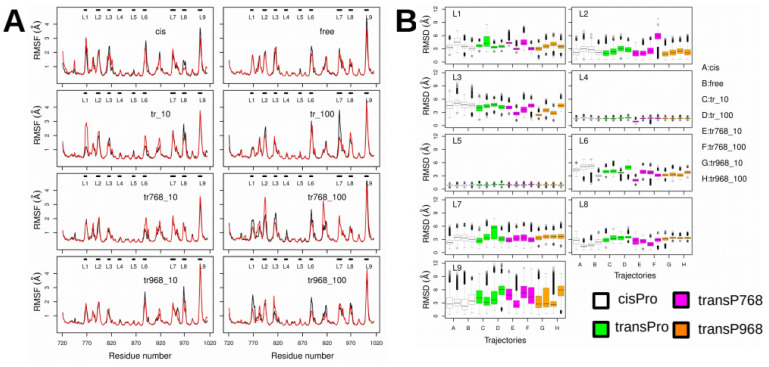
Refined analysis of X-Pro imide bond *cis-trans* isomerization on the internal dynamics of the upper loops. (**A**) Coordinate RMSFs (Å) of the backbone heavy atoms. For each trajectory, the black and red curves correspond to the replicas R1 and R2. (**B**) Distribution of coordinate RMSDs (Å) of the backbone heavy atoms calculated for loops L1 (residues 764–768), L2 (residues 789–795), L3 (residues 812–816), L4 (residues 833–838), L5 (residues 862–866), L6 (residues 884–889), L7 (residues 940–948), L8 (residues 964–970) and L9 (residues 996–1002). Before the RMSDs were calculated, the entire CNF1 structure was first superimposed on the structure taken from PDB entry 1HQ0. The RMSD distributions are plotted as box plots colored white (trajectories cis, free), green (trajectories tr_10, tr_100), magenta (trajectories tr768_10, tr768_100), and orange (trajectories tr968_10, tr968_100). For each trajectory, the distributions of the two replicas are drawn.

**Figure 4 ijms-22-10129-f004:**
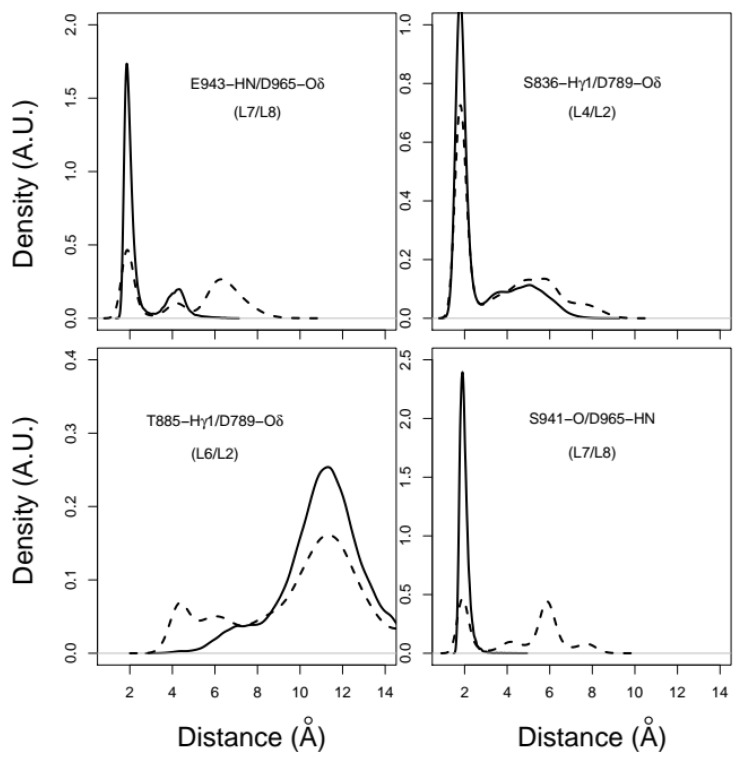
Interloop distance distribution between the following atoms: the HN atom from E943 (L7) and the Oδ atoms of D965 (L8), the Hγ atom from S836 (L4) and Oδ atoms of D789 (L2), the Hγ1 atom from T885 (L6) and Oδ atoms of D789 (L2), and O atom from S941 (L7) and the HN atom from D965 (L8). For distances involving atoms Oδ1/Oδ2, the minimum distance was conserved for each trajectory frame, and the atom was labeled Oδ. The distributions are given for trajectories with *cis* conformations of the X-Pro imide bond (solid line) and for trajectories with *trans* conformations of X-Pro imide bonds (dashed line).

**Figure 5 ijms-22-10129-f005:**
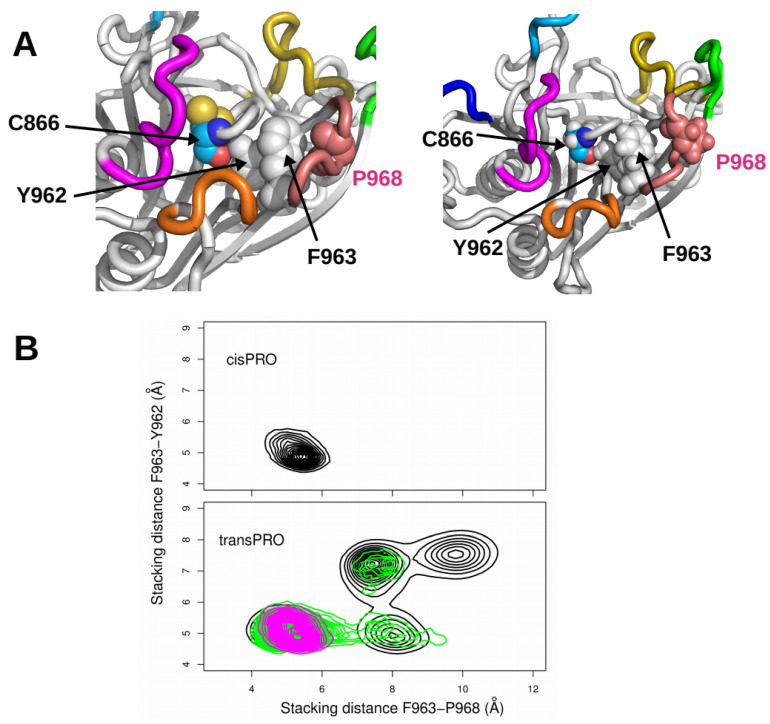
(**A**) Relative positions of residues P968, F963, Y962 and C866 in the X-ray crystallographic structure 1HQ0 (left) and in the last frame of replica R1 of tr_100 (right). (**B**) Plots of the stacking distance between F963 and Y962 (Å) with respect to the stacking distance between F963 and P968 (Å). The stacking distance is defined as the average distance between the atoms belonging to the cycles of these residues. The plots were realized for all trajectories with *cis* conformations of X-Pro imide bonds (top plot) and for all trajectories with *trans* conformations of X-Pro imide bonds (bottom plot): tr100 (black), tr768_100 (magenta), and tr968_100 (green).

**Figure 6 ijms-22-10129-f006:**
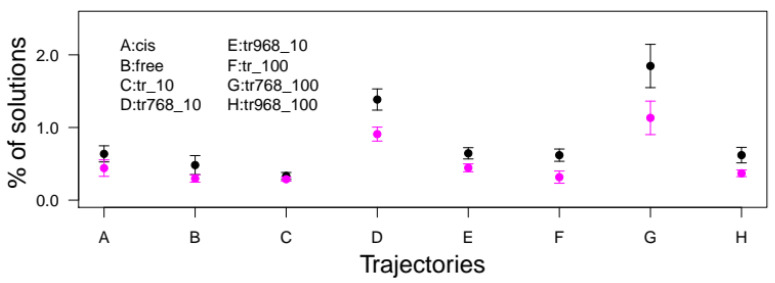
Numbers of FlexPepDock [20] solutions for the representative conformations extracted from different trajectories (see Appendix A). The number of solutions was normalized with respect to the number of docking trials and scaled by 10,000. These numbers of trials were 525,000 (cis), 595,000 (free), 665,000 (tr_10), 525,000 (tr768_10), 665,000 (tr968_10), 420,000 (tr_100), 280,000 (tr768_100), and 735,000 (tr968_100). The results are colored in black for the RhoA peptide and magenta for the Ras peptide.

**Figure 7 ijms-22-10129-f007:**
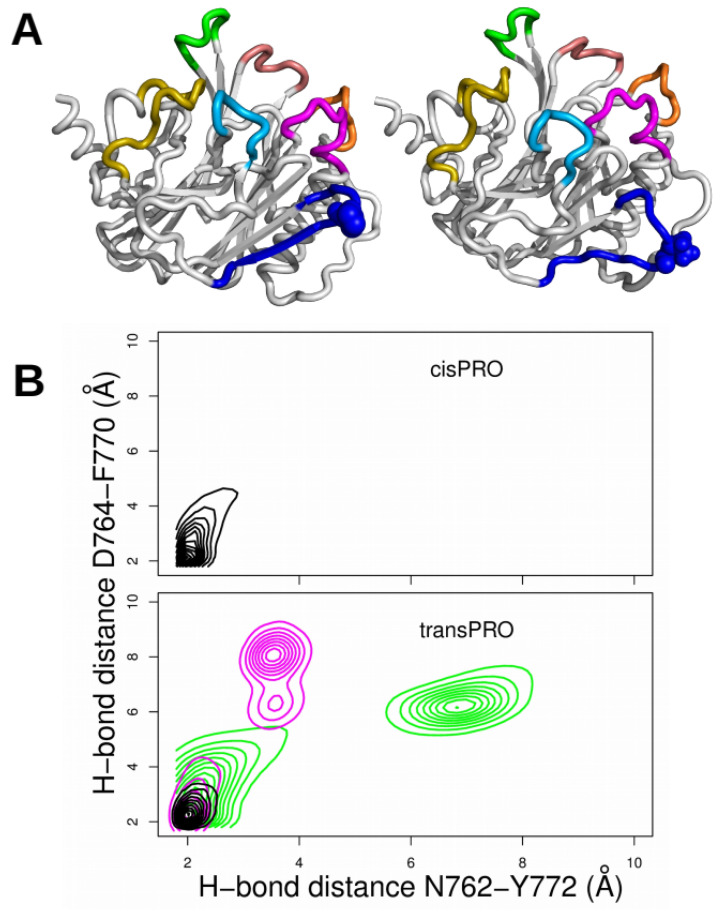
(**A**) Comparison between the 1HQ0 PDB structure (left) and a conformation from the tr768_100 trajectory (right). The loops are colored as in Figure 1. P768 is drawn in spheres and colored blue, and the region 769–775, close to loop L1, is also colored blue. (**B**) Plots of the hydrogen bond distance between D764 and F770 (Å) with respect to the hydrogen bond distance between N762 and Y772 (Å). The hydrogen bond distances are defined as distances between the carbonyl oxygen of F770 and the amide hydrogen of D764 and between the backbone carbonyl oxygen of N762 and the amide hydrogen of Y772. The plots were realized for all trajectories with *cis* conformations of X-Pro imide bonds (top plot) and for all trajectories with *trans* conformations of X-Pro imide bonds (bottom plot): tr100 (black), tr768_100 (magenta), tr968_100 (green).

**Figure 8 ijms-22-10129-f008:**
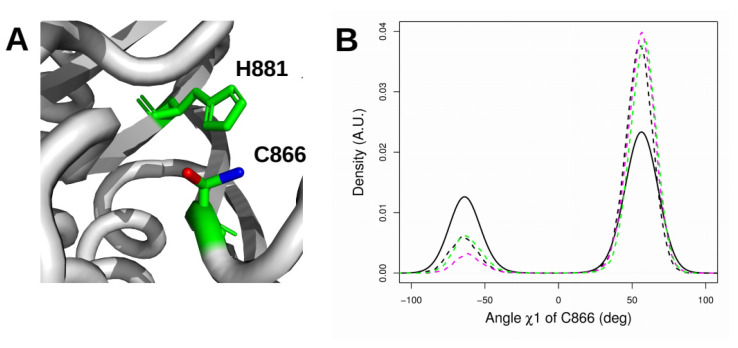
(**A**) Active (red) and resting (blue) orientations of C866 in the PDB structure 1HQ0. (**B**) Distribution of the dihedral angle χ1 (N-Cα-Cβ-Sγ) (°) of C866 for various values of the X-Pro peptidic angle ω (°) for P768 and P968 along MD trajectories. The solid and dashed curves correspond to the trajectories with *cis* and *trans* X-Pro imide bonds, respectively. In the MD simulations the active orientation appears with a dehydral angle value of 60° (40.6° in the crystal structure. The dashed curves are colored according to the *trans* isomers of prolines: *trans*-P768 (magenta), *trans*-P968 (green), and both *trans* (black).

**Table 1 ijms-22-10129-t001:** Characteristics of MD trajectories.

Trajectory	Duration (ns)	Restrained Residues	Restraint Type	Force Constant (kcal/(mol.degrees))
cis	2 × 100	P978, P768	cis	10
free	2 × 100	-	-	-
tr_100	2 × 100	P978, P768	trans	100
tr768_100	2 × 100	P768	trans	100
tr968_100	2 × 100	P978	trans	100
tr_10	2 × 100	P978, P768	trans	10
tr768_10	2 × 100	P768	trans	10
tr968_10	2 × 100	P978	trans	10

**Table 2 ijms-22-10129-t002:** Sequences of peptides docked on CNF1 conformations using FlexPepDock [20]. The Q63 residues undergoing deamidation are labeled with an asterisk (*).

RhoA 60–72	Nter-T A G Q* E D Y D R L R P L
Ras 60–72	Nter-T A G Q* E E Y S A M R D Q

**Table 3 ijms-22-10129-t003:** System composition.

System	Number of TIP3P Waters	Neutralizing Ion	Total Number of Atoms
CNF1 WT	13,160	One Na^+^ ion	44,065

## Data Availability

10.5281/zenodo.5513049.

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
