# Peer review of "Conformational Insights into the Control of CNF1 Toxin Activity by Peptidyl-Prolyl Isomerization: A Molecular Dynamics Perspective"

_ijms, 2021, doi:10.3390/ijms221810129_

Round 1
Reviewer 1 Report
This is an interesting paper about the molecular activation of the catalytic part of the bacterial toxin CNF1. The authors identified two proline residues present in the catalytic domain of the toxin to be important for activity. By peptidyl-prolyl isomerization the prolines P768 and P968 are involved in stabilization of the catalytic cysteine (by stacking interactions) and accessability of the RhoA switch 2 region for deamidation.
The paper is well written and the message clearly presented.
I have only a few minor points which have to be addressed:
1. Figure 3 legend: do you mean replicas R1 and R2?
2. Citation of figures is labeled with errors throughout the paper.
3. What about the third amino acid of the catalytic triade of CNFs, please comment within the discussion.
Author Response
see the attached pdf file containing the reply to all reviewers.

Reviewer 2 Report
Paillares and co-workers performed in silico studies on CNF1 toxin and the role of the exposure two PRO residues on secondary/tertiary structures and mobility of the catalytic entrance loops. Although I consider the manuscript theme interesting, as the isomerization of only 2 proline amino acids can cause important changes in function and mobility of loops, sometimes the paper is hard to follow and could gain with some modifications.
Below are my detailed comments:
- In Introduction section, right in the 1st paragraph, contextualize the role of CNF1 in diseases (uropathogenic, meningitis, other), explaining why this problematic is important to address.
- In the 2nd paragraph, please remember the readers about the basic proline structural characteristics. Why is this amino acid unique? Why are you looking at the omega angle? Why proline impacts the global protein structure?
As you must know Proline is the only residue that can be found in cis conformation, thus a peptide bond will only adopt a cis or trans conformation in a bond that precede a proline residue, which is slightly different of what you stated.
- In results, did you consider calculating the SASA for both prolines, in both forms (cis/trans)? This could be useful to associate the proline structural behavior with some hydrophobic/hydrophilic effect felt by the toxin, which can interfere in the global 3D organization.
- There is no consistency in Figures’ presentation. While in Figure 1, the letter A/B are in a reasonable size, in Figure 2 you used a) and f) form, in Figure 3, 7 and 8, the A/B is very large...
- Please, in the first time you mentioned “stacked interactions” refer that although proline is not an aromatic residue, can perform stacking with aromatic amino acids, especially when in trans conformation. doi: 1021/ar300087y
- The Methods section is mostly well written and the computational protocol is clear. In point 4.6, mentioning the FlexPepDock program is not sufficient to infer that you optioned for a flexible docking, this should be stated. Also, explain why is sufficient to use two opposite initial orientations, manually positioned, not implying any bias.
Author Response
see the attached file for reply to all reviewers.

Reviewer 3 Report
Introduction
The introduction is concisely written and provides sufficient information for readers to understand the aim of the study. However, I think it would help if the authors could clarify why CNF1CD was chosen for some of the experiments instead of the wild-type protein.
Results
Most of the embedded figure references/links are displayed as error messages.
Figure 1: The text states that the structure of CNF1CD is depicted, but the figure legend says CNF1. The number of replicates for the Western analysis of RhoA deamidation (1C) and RhoA degradation (1D) is not mentioned in the legend or the method section.
Figure 3: This figure is a bit small, and the lettering is slightly blurred. Line 204, the duplicates are named R1 and R1.
In general, I found it quite a challenge to distinguish when and why CNF1/CNF1CD was used. For example, Figure 1: Why was CNF1CD used for the analysis of the deamidase activity but the wild-type variant (and its proline-substituted mutants) for the evaluation of RhoA degradation?
Discussion
I could not follow the reasoning in line 363ff. CNF1CD carries a cysteine to serine substitution at position 866 (Introduction, line 78), how would the thiol residue then play a role in the orientation switch of the protein. I also do not get your last hypothesis stating that isomerisation of P768 favours RhoA-CNF1 interaction based on the observations using CNF1CD (line 434); I would appreciate it if you could explain that further.
Material & Methods
Methods are adequately described. Line 519f seems to be missing a word.
Supplementary Information
S1: "R1 and R1"
Minor Points
Abstract, line 21: "[…] being easier than for P968 […]"
Line 64: "[…] thereby conferring to these proteins gain-of-function properties."
Line 68: Typo "show"
Line 159: Reference
There are double punctuation marks at the end of some figure legends.
Author Response

(The authors gave the same response as above.)

Round 2
Reviewer 2 Report
The authors responded to all questions and suggestions, and improvements were made throughout the manuscript.
Please, next time, pay attention to the lines referred to in the rebutal letter, as they do not match the some of the modifications made in the paper, so I had to look for some new phrases.